# Chest X-ray Interpretation: Detecting Devices and Device-Related Complications

**DOI:** 10.3390/diagnostics13040599

**Published:** 2023-02-06

**Authors:** Marco Gambato, Nicola Scotti, Giacomo Borsari, Jacopo Zambon Bertoja, Joseph-Domenico Gabrieli, Alessandro De Cassai, Giacomo Cester, Paolo Navalesi, Emilio Quaia, Francesco Causin

**Affiliations:** 1Institute of Radiology, Department of Medicine (DIMED), University of Padova, 35121 Padua, Italy; 2Department of Neuroradiology, University Hospital of Padova, 35121 Padua, Italy; 3Anesthesia and Intensive Care Unit, University Hospital of Padova, 35121 Padua, Italy; 4Anesthesia and Intensive Care Unit, Department of Medicine (DIMED), University of Padova, 35121 Padua, Italy; 5Institute of Radiology, University Hospital of Padova, 35121 Padua, Italy

**Keywords:** CXR, malposition, CVC, pneumothorax, device, complications

## Abstract

This short review has the aim of helping the radiologist to identify medical devices when interpreting a chest X-ray, as well as looking for their most commonly detectable complications. Nowadays, many different medical devices are used, often together, especially in critical patients. It is important for the radiologist to know what to look for and to remember the technical factors that need to be considered when checking each device’s positioning.

## 1. Introduction

The chest radiograph, or chest X-ray (CXR), is a well-established imaging modality that is widely used to check for lung, pleural and mediastinal abnormalities [1]. In addition, it is also a first-line examination to monitor the correct positioning of medical devices.

A CXR is usually performed to check for immediate and late-onset complications, even before they can potentially become clinically significant (e.g., the early identification of a small pneumothorax can prevent it from degenerating into a tension pneumothorax).

Potential confounding factors are EKGs’ leads, external tubes, artifacts and overlapping of different devices. The need to have at least two orthogonal projections to correctly locate a device can be a limit in patients in which only an anteroposterior projection is feasible. Moreover, technical factors must be considered, such as the orientation of the X-ray tube and patient rotation (Figure 1) [2]. 

## 2. Classification of Medical Devices

In this article, medical devices are disclosed according to their anatomical location: vascular devices (central vein catheter, peripherally inserted central catheter, port, midline and Swan-Ganz catheter), airway devices (endotracheal tubes, tracheostomy tubes), cardiac devices (loop recorder, pacemaker, automatic implantable cardioverter-defibrillator, ventricular assistance devices, Impella, intra-aortic balloon pump and extracorporeal membrane oxygenation), gastro-intestinal devices (nasogastric tube and nasoenteric tubes), chest tubes and miscellaneous other devices that detectable upon CXR (Table 1). 

All X-ray images have been anonymized.

## 3. Intravascular Devices

### 3.1. Central Vein Catheter (CVC)

CVC refers to a wide range of catheters that are used to obtain reliable central venous access in patients needing prolonged intravenous therapy or in critically ill patients. There are a variety of these catheters, varying in size and shape, but on a CXR, they all appear radiopaque and can be easily detected [3]. 

CVCs are usually inserted from the jugular or subclavian vein, both on the left or right side (they can also be introduced from femoral veins, but this review considers only the thoracic location) (Figure 2) [4].

The tip of the catheter should be positioned into the superior vena cava (SVC) at the superior cavoatrial junction or in the right atrium [5].

The most common CXR-detectable complications are pneumothorax, malposition (looping, inferior vena cava (IVC) positioning of the tip, wrong thoracic vein positioning), or, less often, vascular perforation, with the catheter located inside the thorax (Figure 3, Figure 4 and Figure 5, Table 2) [6,7]. 

If there are any doubts about vascular perforation, chest CT should be performed before removing the catheter to assess its exact location [8]. CXR is not able to diagnose catheter-related infections or catheter thrombosis. Arterial puncture is almost always diagnosed by the operator by seeing bright red and pulsating blood in the cannula.

As previously mentioned, technical factors such as the rotation of the patient should be considered, because they can alter the projection of the catheter, a potential confounding factor leading to the wrong diagnosis of a displaced device. 

### 3.2. Peripherally Inserted Central Catheter (PICC)

PICC is a venous catheter that is used for medium- and long-term medical therapy administration. It is a central catheter, but a peripheral arm vein is used as an insertion point. It is usually seen in CXR arising from the left or right arm, with the ideal tip position at the level of the SVC or at the cavoatrial junction (Figure 6). Due to the peripheral insertion point, PICC is usually less associated with the iatrogenic pneumothorax. However, if forced against the vessel wall, the PICC can still cause pneumothorax or hemothorax (Table 2) [9]. 

### 3.3. PORT, or Totally Implantable Vascular Access Device

A PORT is an infusion device made of a catheter and a small chamber placed beneath the skin into which drugs can be injected through a silicone membrane. It allows for long-term intravenous therapy, and it is specially used in hematology and oncology patients. The PORT can be seen on a CXR against pectoral tissues with a catheter arising from a small radiopaque device, usually round-shaped, placed via the jugular or subclavian vein, with its tip ending inside the SVC or in the right atrium (Figure 7). The most common CXR-diagnosable complication of port implantation is pneumothorax (Figure 8, Table 2) [10]. 

### 3.4. Midline

A midline is a mid- and long-term peripheral venous access catheter. It is usually inserted from the brachial vein, and the tip is usually placed in the axillary or subclavian vein. In a normal CXR, only the tip of the catheter can be evaluated. 

It is important to know whether the patient has a midline or a PICC, because on a CXR, a midline can appear just like a mispositioned PICC catheter, with the CXR being able to study just the distal part of it. 

Considering the location of the midline, the possibility of a pneumothorax as a complication is remote. If the tip of the catheter is too proximal or is misplaced in a brachial vein, the CXR is not able to provide a diagnosis, because the catheter would be out of the field of view (Table 2).

### 3.5. Swan-Ganz Catheter

The Swan-Ganz catheter is a balloon-tipped catheter that is used to measure pulmonary artery pressure. It is usually inserted from the right jugular vein, making its way to the right atrium, right ventricle and main pulmonary artery. Usually, the tip is wedged into a right divisional artery (Figure 9) [11].

Complications include perforation of the right atrium or of the pulmonary artery wall; these are life-threatening situations that usually require surgical treatment, and the diagnosis is mainly based on clinical features or bedside echocardiographic findings. The CXR has a role in diagnosing looping or knotting of the catheter (Table 2) [6,12]. 

## 4. Airway Devices

### 4.1. Endotracheal Tubes

Endotracheal tubes (ETTs) are essential devices in anesthesiological practice, allowing for the artificial ventilation of the lungs.

They can be single- (SLT) or double (DLT)-lumen tubes, the latter allowing for differential ventilation or anatomic separation of the lungs; they are usually inserted through the oral cavity, although nasal access can be used when the previous method is not viable. 

SLTs usually consist of wide-bore plastic tubes (visible in a plain radiograph through a radiopaque strip within them) with an inflatable balloon at their tip, in order to fix them in place and prevent the aspiration of gastric contents, blood, secretions and other fluids.

In adults, a correctly positioned SLT has its deep extremity placed in the trachea, 5 cm (±2 cm) proximally to the carina (Figure 10); when assessing its position, eventual flexion/extension attitude of the neck should be taken into account (by looking at the mandibula, when included in the radiogram), adjusting the measurement by up to +2/−2 cm, respectively [3].

If the carina is not clearly recognizable, the aortic knob can be taken as a point of reference; the tip of the ETT should be located right above its side [3].

ETT malposition is a relatively common situation following an intubation procedure: it can be placed too high, risking a cord trauma, or too low, selectively cannulating a mainstem bronchus (usually the right one, for anatomic reasons) (Figure 11), and therefore causing atelectasis of the contralateral lung (accompanied by onset/worsening of hypoxemia) and overinflation of the ipsilateral lung (risk of barotrauma and subsequent iatrogenic pneumothorax) [13,14].

Oesophageal intubation should be suspected if an anomalous gaseous distention of the stomach is noted while lungs remain hypo-expanded; in addition to not improving the respiratory function, such a malposition of the ETT can dramatically lead to an oesophageal perforation. If not detected clinically, it is therefore crucial to promptly identify the perforation on plain radiography [15]. 

In this regard, it is important to note that such a misplaced ETT is not always clearly projected outside the tracheal lumen on the AP projection, making it challenging to provide a confident diagnosis. In these situations, an oblique projection can be helpful [16].

The last condition that should be checked in a CXR after ETT positioning is eventual overinflation of the balloon, which can cause throat pain, tissue ischemia and tracheal perforation/fistulae formation (Table 3) [17,18].

### 4.2. Tracheostomy Tubes

Tracheostomy is usually performed when long-term artificial ventilation is needed.

The tip of the tracheostomy tube should be located one-half to two-thirds of the distance from the stoma to the carina (Figure 12), and contrary to the ETT, it does not variate its position with flexion-extension movements of the neck [19].

Complications associated with tracheostomy positioning include subcutaneous emphysema, hematoma and pneumomediastinum (although a small amount of subcutaneous and/or mediastinal air can normally be seen); tracheal stenosis derived from granulation tissue formation and fibrosis may develop at the site of the stoma (Table 3) [3,19].

## 5. Cardiac Devices

A variety of devices used to monitor heart rhythm and for therapeutic aims can be found on a CXR when analyzing the mediastinum. 

### 5.1. Loop Recorder

A loop recorder is a long-term hearth rhythm monitor that is usually seen in a CXR as a small radiopaque device with no catheters or electro stimulator wires against the mediastinum (Figure 13). It is located in the subcutaneous tissues and should not cause any CXR-diagnosable complications (the migration of the loop recorder along the chest wall is anecdotical) [20,21]

### 5.2. Pacemaker (PM) and Automatic Implantable Cardioverter-Defibrillator (AICD)

PMs and AICDs are cardiac conduction devices that are made to monitor and to intervene upon cardiac rhythm. They can be external or internal, usually placed under the skin just above the clavicle, and they have electrical leads running inside the veins towards the heart chambers.

In the case of an external PM, only the stimulator wires can be recognized in a CXR. They are usually inserted from a jugular or femoral vein, or in some cases, directly during heart surgery. Their tips can be recognized as projecting against the right atrium and ventricle.

AICD and PM can be distinguished on a CXR by the presence of ICD shock coils: a thick metal segment placed in the end of the lead. Combined ICD-PM devices also exist, known as “cardiac resynchronization therapy” (Figure 14). 

CXR is useful after the positioning of these devices to check for the correct location of the electrical leads and to check for their integrity on follow-up examinations. Twiddler’s Syndrome, the consequence of the patient manipulating the device in its pouch, can result in misplacement of the wires that are visible on the CXR as twisted around the PM body [22]. 

### 5.3. Ventricular Assistance Devices (VADs) 

VADs are mechanical circulatory support devices that can be implanted both in the right (RVAD) or left (LVAD) ventricle; in some cases, RVAD and LVAD can be placed together. 

These devices are surgically implanted, with a pump communicating with the ventricle through an inflow cannula while an outflow cannula delivers blood to the aorta (in the case of LVAD) (Figure 15) or to the pulmonary artery (in the case of RVAD) [23]. 

A CXR can provide information regarding the correct positioning of the cannula. Bleeding from the insertion point can cause hemopericardium, leading to cardiac tamponade or hemothorax; other possible complications of VADs (thrombosis/venous thromboembolism, cardiac arrhythmias, solid organ dysfunction, driveline infections) are not detectable on a CXR [24,25,26,27].

Frequently, the pump of the VAD covers part of the thorax on a CXR, therefore making it difficult for the radiologist to evaluate lung parenchyma and pleural space in that particular site (Table 4).

### 5.4. Impella 

An Impella is a left ventricular assistance device. It is placed inside the left ventricle through the aortic valve, and it pumps blood from the left ventricle into the ascending aorta (Figure 16). Principal complications are related to the arterial femoral access, and an Impella is usually placed using radioscopy, therefore reducing the risk of mispositioning [25,28]; nevertheless, cases of Impella dislocation are reported in literature [29] (Table 4).

### 5.5. Intra-Aortic Balloon Pump (IABP)

An intra-aortic balloon pump is an inflatable device that improves perfusion of the coronary arteries.

An IAPB catheter is radiolucent, except for two radiopaque marks at both endings to allow for its identification on CXR. The ideal positioning of the IABP should be at the level of the descending aorta, below the origin of the left subclavian artery and above the splanchnic vessels, to avoid complications such as the occlusion of these vessels or losing of device functionality. On a CXR, an IABP should be at the aortopulmonary level (Figure 17).

The main complications of IABPs include aortic dissection and malpositioning. The CXR has a role in diagnosing mainly the latter, the first complication being a clinical emergency that needs an emergency CT angiography (Table 4) [30].

### 5.6. Extracorporeal Membrane Oxygenation (ECMO)

ECMO includes a series of bypass techniques that allow for blood oxygenation in patients with severe respiratory or cardiocirculatory failure [31]. 

Traditional circuits include veno-venous (VV) and veno-arterial (VA) ECMO [31].

The VV-ECMO is used mainly in the setting of severe isolated respiratory failure, draining blood from a cannula usually placed in the SVC and pumping it back into a peripheral vein (usually the femoral vein); the patient’s own heart then pumps the blood throughout the body (Figure 14) [31,32,33].

The VA-ECMO, on the other hand, is the preferred modality in patients with isolated cardiac or combined cardiopulmonary failure. It drains deoxygenated blood from a large vein (usually the femoral vein or the IVC) and reintroduces it into a large artery (usually the subclavian or the femoral artery) after oxygenation, therefore bypassing the heart [31,33].

Even though a CXR is not indicated to monitor the correct functioning of the ECMO circuit, it can still be useful in recognizing the position of ECMO cannulas, considering their variable configuration according to the type of ECMO and the kind of vascular access performed [34,35]. 

A suspicion of misplacement or migration can be raised according to CXR and should be confirmed via CT angiography (Table 4) [36].

### 5.7. Other Cardiac Devices

The following section discusses a series of CXRs depicting other cardiac devices that are not included in the previously mentioned ones (Figure 18, Figure 19 and Figure 20).

## 6. Gastro-Intestinal Devices

### 6.1. Nasogastric Tubes

Nasogastric intubation is a medical process involving the insertion of a radiopaque-marked plastic tube (nasogastric tube or NG tube) through the nose (sometimes through the oral cavity, referred to in this case as an orogastric tube) to reach down through the stomach.

The NG tube has a double function: it can be used for both the administration of enteral nutrition/oral drugs and for the drainage of gastric contents [37].

A properly positioned NG tube descends into the thorax and crosses the diaphragm on the midline, ending about 10 cm below the oesophago-gastric junction (Figure 21) [38]; note that often, an abdomen X-ray may be additionally required in order to define the abdominal position of the tube.

NG tube malposition includes a tip that is placed too high, mostly in the distal oesophagus (Figure 22) (note that in case of a *pH probe*, this is considered the adequate placement) or pushed too low (Figure 23), especially in the duodenum (where it could be harmful) (Figure 24); there could also be coiling in the upper ways (Figure 25) or kinking in the gastric lumen (Figure 26) [19,39,40].

Insertion into the respiratory tree (Figure 27) (tip in the trachea/mainstem bronchus/lobar bronchus) can lead to subsequent aspiration pneumonia [37].

Breaching of the gastric wall can be easily recognized if the tip lies in the abdominal cavity, and signs of perforation (i.e., subdiaphragmatic free gas) are noticeable [19].

Intracranial insertion of the NG tube is rare, mostly reported in cases of skull base trauma or surgery (Table 5) [41].

### 6.2. Nasoenteric Tubes

Nasoenteric tubes are small-bore plastic tubes that are used when prolonged artificial feeding is needed (especially Dobhoff tubes); they are not capable of suction.

Considering that the most correct location of their tip is in the second portion of the duodenum [19], one should be aware of their existence in order to avoid mistaking them for mispositioned nasogastric tubes (Table 5).

## 7. Chest Tubes

The insertion of tubes through the chest wall into the pleural space (tube thoracostomy) is a well-assessed practice in the hospital setting, allowing for both drainage of collected air/fluid (pneumothorax, pleural effusion, hemothorax, cardio-thoracic surgery, etc.) and drug administration (antibiotics, saline, chemical pleurodesis) [42].

Chest tubes include a wide variety of sizes (small- and large-bore chest tubes) and shapes (straight, angled, pig-tailed), all provided with a radiopaque strip presenting a small gap in correspondence of the first side-hole [42].

The proper position of the tube depends on its aim, an antero-apical orientation being recommended for drainage of the pneumothorax (Figure 28 and Figure 29); conversely, in the case of evacuation of a pleural effusion, the tip should lie in the inferior-posterior region (Figure 30 and Figure 31) [3].

Kinking and malpositioning of the tube are not infrequently seen and should always be suspected in the case of ineffective pleural drainage [19].

The most common sites for misplaced chest tubes include extrapleural soft tissues, intrafissural/intraparenchymal positioning and juxtaposition against the mediastinum. Meanwhile, though inadvertent advancement of the chest tube into the mediastinum appears to be uncommon, there are reports of iatrogenic placement of the chest tube through the diaphragm into the abdomen, causing laceration of the liver, spleen or stomach [19,43,44,45,46].

Intermittent obstruction of chest tubes is usually attributable to clotting blood, debris or pus; however, such abnormalities are not directly visible in a CXR.

Once a chest tube is removed, a residual thickened pleural or parenchymal line may be noticed along its previous tract on CXR. This is not a pathological finding and should not be misinterpreted as a pneumothorax (Table 6) [19].

## 8. Miscellaneous

### 8.1. Cerebrospinal Fluid (CSF) Shunts

The main indication of CSF shunts is the treatment of hydrocephalus.

Whereas the proximal catheter is usually inserted intp one of the lateral ventricles, the distal catheter can theoretically be placed in any fluid-reabsorbing body cavity, peritoneum being the preferential location (ventriculoperitoneal or VP shunt) (Figure 32), as it is associated with fewer complications. The right atrium (ventriculoatrial or VA shunt) or pleural space (ventriculopleural shunt) are also possible [3].

Complications that are visible on plain radiographs include breakages (Figure 33), disconnections and migrations of the distal catheter, pneumothorax, subcutaneous emphysema and, in VA shunts, features of pulmonary hypertension (Table 7) [3,47].

### 8.2. Vagal Nerve Stimulator (VNS)

VNS are used for long-term management of seizures in refractory patients.

On a CXR, they resemble a pacemaker or ICD, differentiating themselves from these by the fact that the lead is positioned in the neck to stimulate the left vagal nerve in the carotid sheath [3].

### 8.3. Other Devices

The following section presents a series of CXRs depicting various devices that are not included in the previous descriptions, which can also be found on a plain chest radiograph (Figure 34, Figure 35, Figure 36, Figure 37, Figure 38, Figure 39 and Figure 40).

Note that in some cases, abdominal devices can be seen on a CXR (Figure 37, Figure 38, Figure 39 and Figure 40), especially if the radiogram is extended to the upper abdomen.

## 9. Deep-Learning-Based Algorithms for Chest Devices’ Detection on CXR: A Glimpse into the Future

The use of artificial intelligence (AI) software has already demonstrated improved sensibility, specificity and accuracy if used as a decision support for the radiologist when evaluating CXR for thoracic pathologies; other algorithms, on the other hand, have proven to help the radiologist when evaluating follow-up X-rays of the same patient [48,49].

The accurate identification of chest devices by AI in the context of a hypothetical CXR-automated reporting software presents two main purposes, the first being a better evaluation of eventual concomitant chest pathologies, obtainable with the reduction of false positives represented by devices opacities [50]; on the other hand, when the assessment of the location of chest devices via CXR is necessary for many patients at a time (as is often the case in the clinical care context of a hospital setting), a high-sensitivity deep-learning-based algorithm could potentially reduce the time required for the diagnosis of any misplacement, allowing for a more immediate solution and helping to decrease possible misinterpretations [9].

AI algorithms can also be used to monitor potential device-related thoracic complications, such as pneumothorax or pleural effusion [51,52].

Even though a few examples of deep-learning systems for the automatic detection of chest devices have already been proposed [9,50,53,54,55,56], further studies and efforts are necessary in order to provide a comprehensive software tool that can be concretely and reliably applied in the variegated clinical scenario of the hospital reality.

## 10. Conclusions

Multiple chest- and non-chest-related devices can be seen on a CXR, the use of some of which has only recently been introduced in the hospital setting. It is important for the radiologist to know the normal appearances of such devices as well as their proper location, in order to promptly recognize any mispositioning and correctly address clinical management before any major complications develop.

The automated detection of chest devices through deep-learning-based algorithms will hopefully not be only a helpful tool in this scope for the radiologist, but will also improve AI sensitivity and specificity in the assessment of chest pathologies through CXR examination.

## Figures and Tables

**Figure 1 diagnostics-13-00599-f001:**
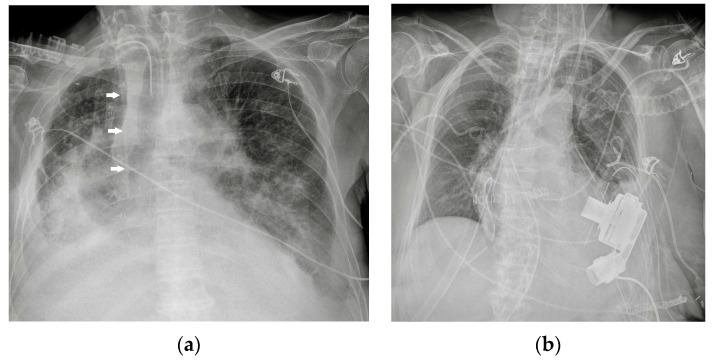
Potential confounding factors when checking for devices on a CXR. (**a**) Anteroposterior CXR of a right-sided rotated patient can simulate the dislocation of the central venous catheter (white arrows). (**b**) Anteroposterior CXR of a patient with multiple devices and external EKG leads.

**Figure 2 diagnostics-13-00599-f002:**
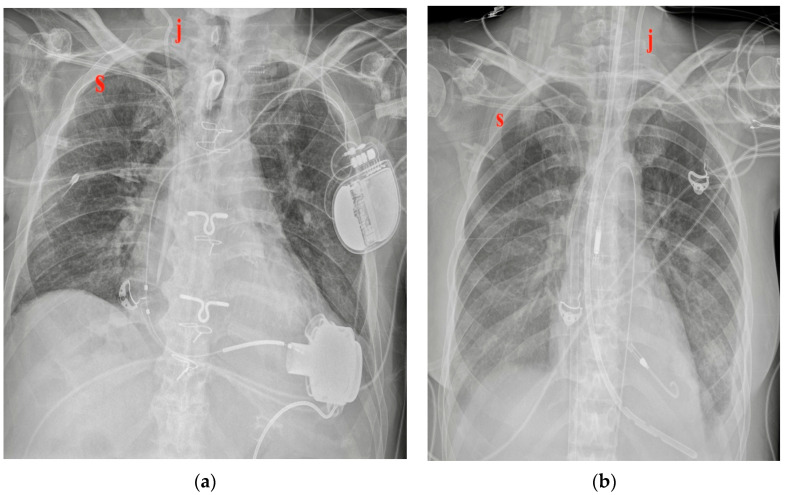
Two different CVC insertion points. (**a**) Right jugular CVC (marked “j”) seen as arising from the neck of the patient, together with a right subclavian CVC (marked “s” in red); note that the two tips of the catheters tend to overlap. (**b**) Subclavian right CVC (“s” in red) and a left jugular CVC (“j” in red).

**Figure 3 diagnostics-13-00599-f003:**
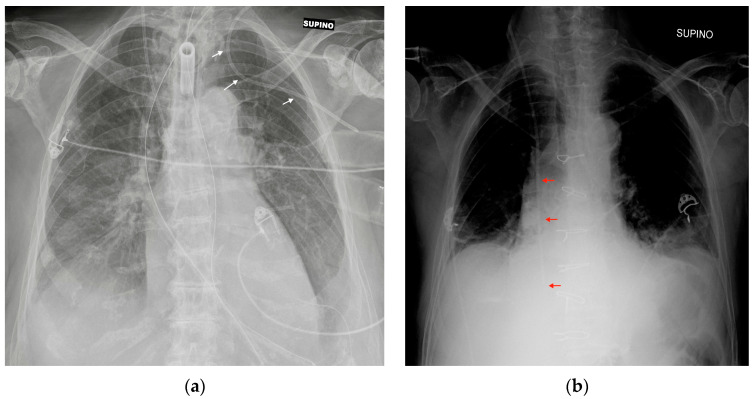
Mispositioned CVCs. (**a**) Left jugular-inserted CVC with its tip inside the left axillar vein (white arrows). (**b**) Right jugular CVC with its tip inside IVC (red arrows). Note that due to underexposure during acquisition of the CXR, the tip of the catheter is hard to see in the upper abdomen.

**Figure 4 diagnostics-13-00599-f004:**
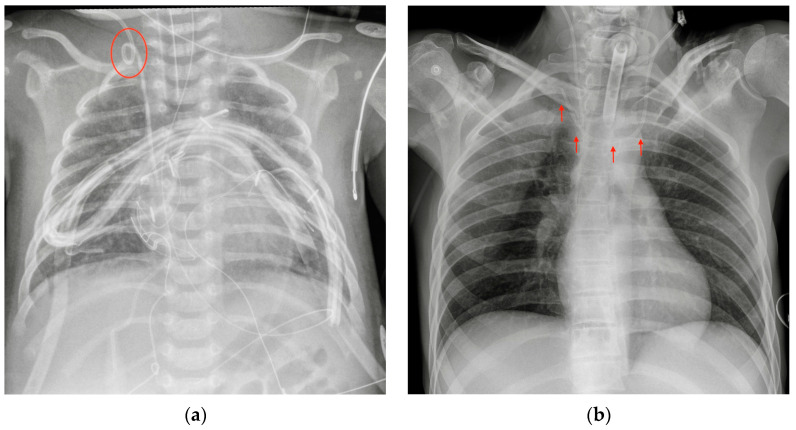
Malpositioned CVCs. (**a**) Right jugular-inserted CVC looping against the vessel wall and ending inside the SVC (red circle). (**b**) Left jugular CVC with tip ending inside the right subclavian vein (red arrows).

**Figure 5 diagnostics-13-00599-f005:**
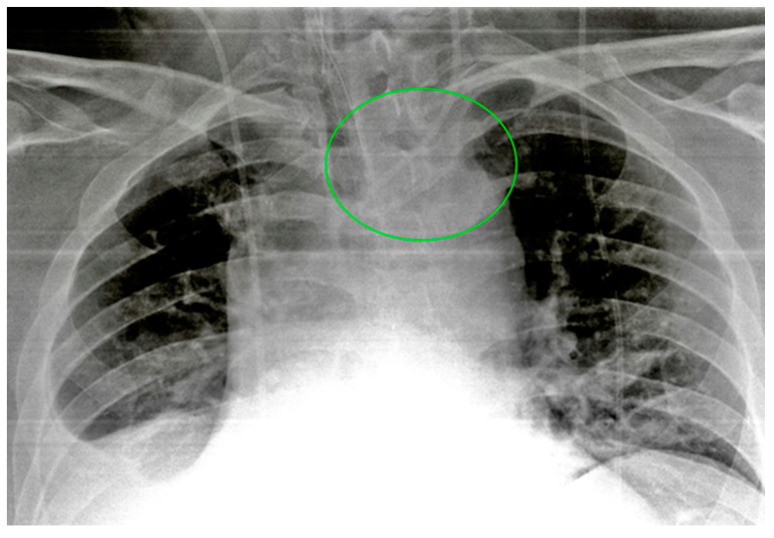
A detail of a CXR showing a left jugular CVC looping with tip ending in the left jugular vein (green circle). Note that due to the underexposure of the CXR, the device is difficult to detect.

**Figure 6 diagnostics-13-00599-f006:**
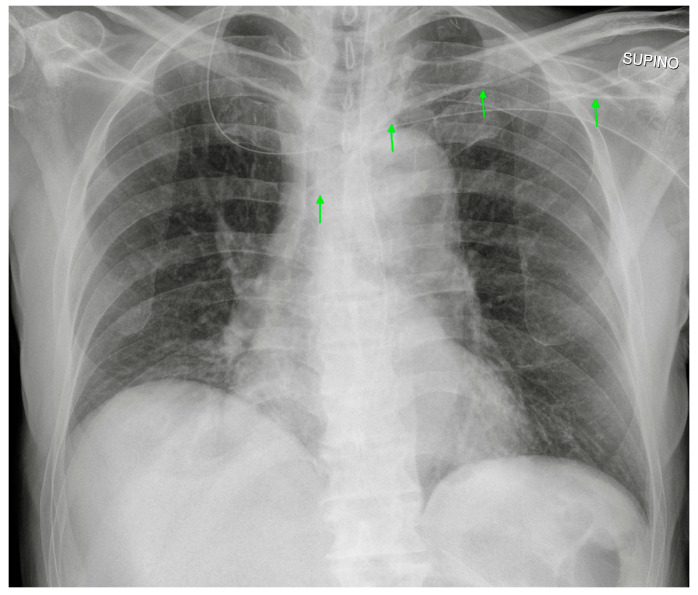
Left-inserted PICC with tip inside the SVC (green arrows).

**Figure 7 diagnostics-13-00599-f007:**
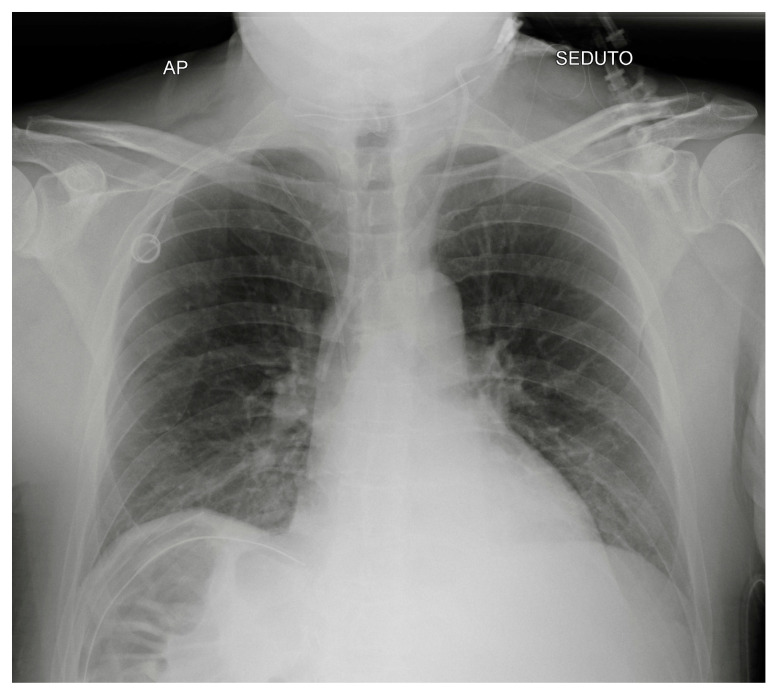
A normally placed PORT catheter with tip at the level of the SVC. Note a left jugular CVC with tip at the same level.

**Figure 8 diagnostics-13-00599-f008:**
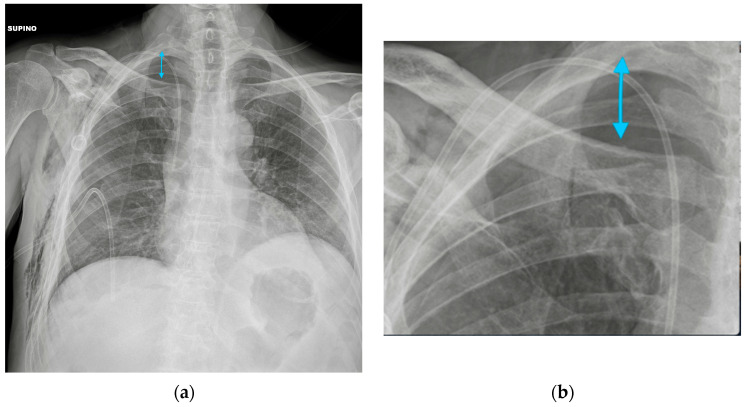
Pneumothorax after PORT positioning. (**a**) A right apical pneumothorax (blue arrow) after the insertion of a right PORT device. A right pleural tube has been placed after the diagnosis. Note the right soft-tissue emphysema. (**b**) A detail of the right apex of the same CXR showing the pleural line.

**Figure 9 diagnostics-13-00599-f009:**
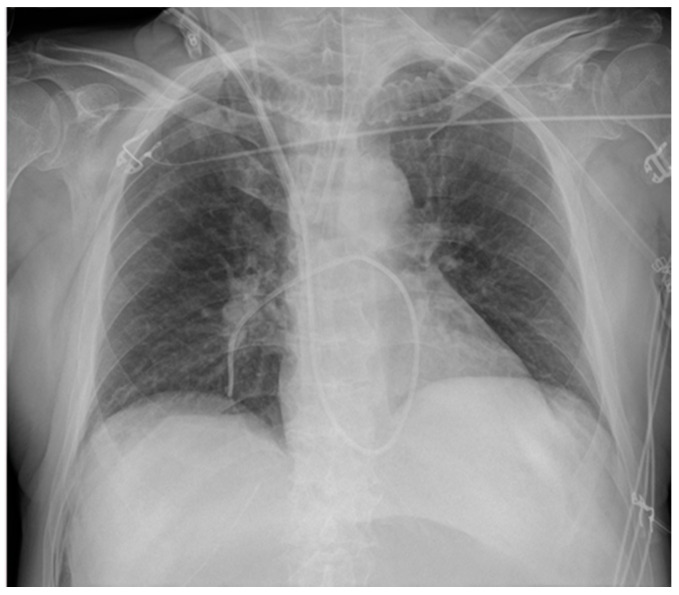
A Swan-Ganz catheter with the tip inside a right lower lobe pulmonary artery branch.

**Figure 10 diagnostics-13-00599-f010:**
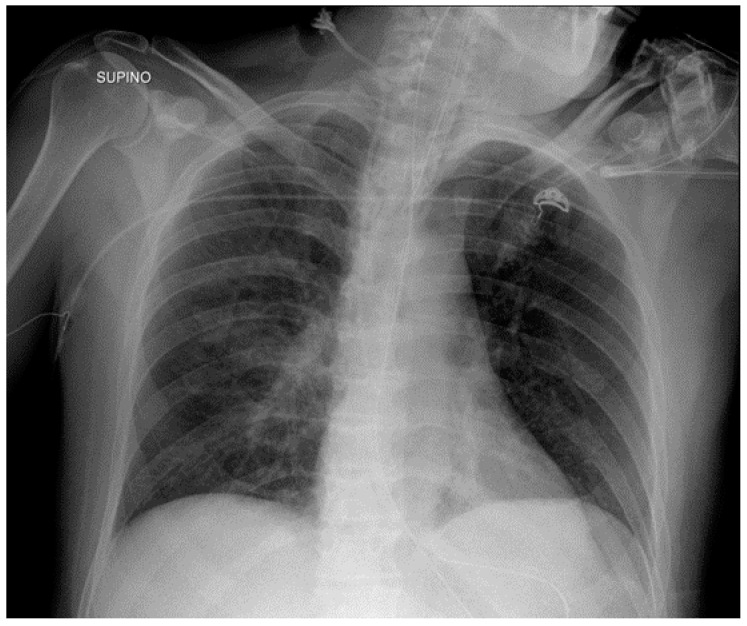
Image of a well-positioned ETT: tip ~5 cm from the carina (right above the aortic knob), no balloon overinflation. In the radiogram, an adequately positioned nasogastric tube whose tip lies in the cardial region of the stomach is visible, after looping along the greater curvature.

**Figure 11 diagnostics-13-00599-f011:**
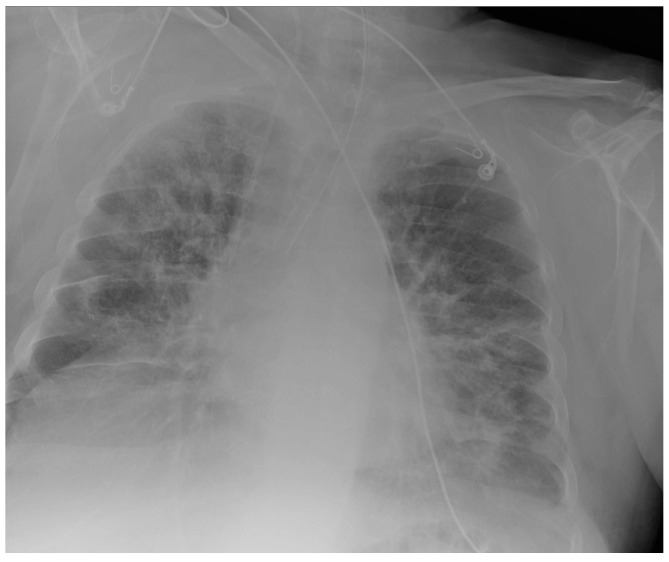
Image of mispositioned ETT whose tip has been pushed too deeply into the right mainstem bronchus. This CXR also shows a right jugular CVC.

**Figure 12 diagnostics-13-00599-f012:**
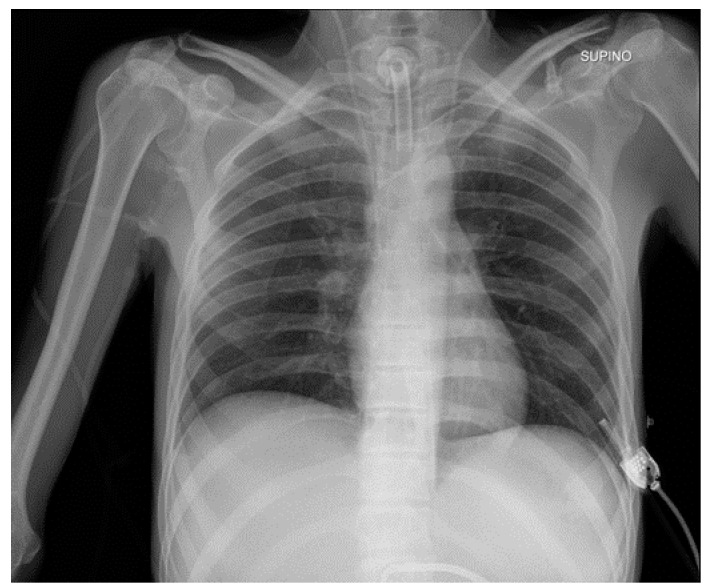
Image of a properly positioned tracheostomy tube (tip at one-half the distance from the stoma to the carina); the CXR also shows right and left jugular CVCs.

**Figure 13 diagnostics-13-00599-f013:**
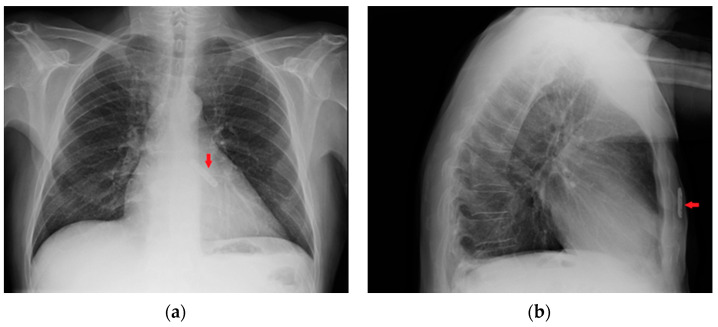
Images of a subcutaneous implantable loop recorder on PA (**a**) and LL (**b**) views of a CXR (red arrows).

**Figure 14 diagnostics-13-00599-f014:**
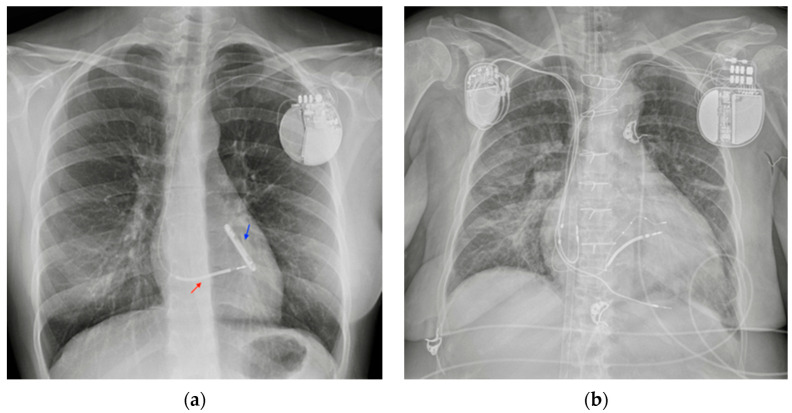
PM and AICD. (**a**) A normally inserted AICD. The thick coil can easily be seen at the end of the lead (red arrow). This patient also has an implantable loop recorder (blue arrow). (**b**) A patient with both a PM (right side) and an ICD–biventricular pacemaker combination (left side); also in the image, a right jugular CVC, an ETT and an extracorporeal membrane oxygenation (ECMO) cannula whose tip reaches the right atrium via the IVC (femoral access) (Table 4).

**Figure 15 diagnostics-13-00599-f015:**
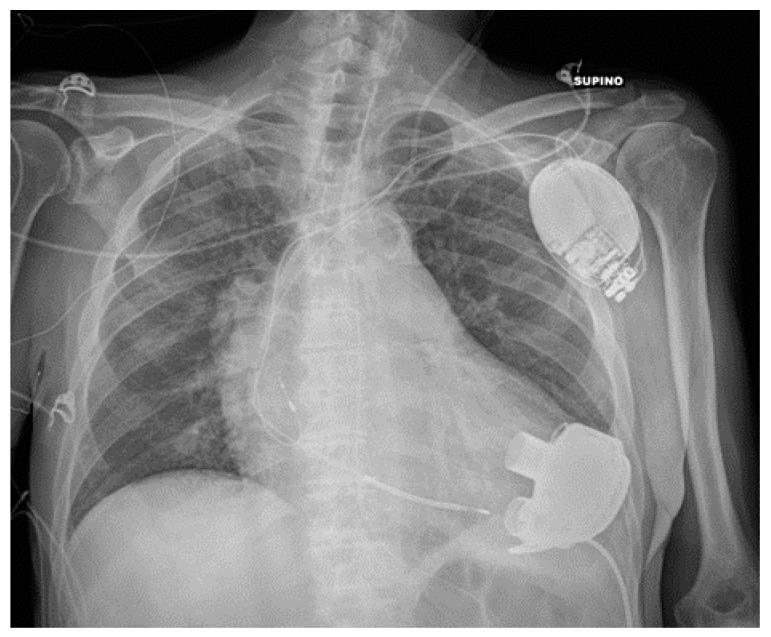
Properly positioned LVAD. The image also shows an AICD, a left jugular CVC and a nasogastric tube.

**Figure 16 diagnostics-13-00599-f016:**
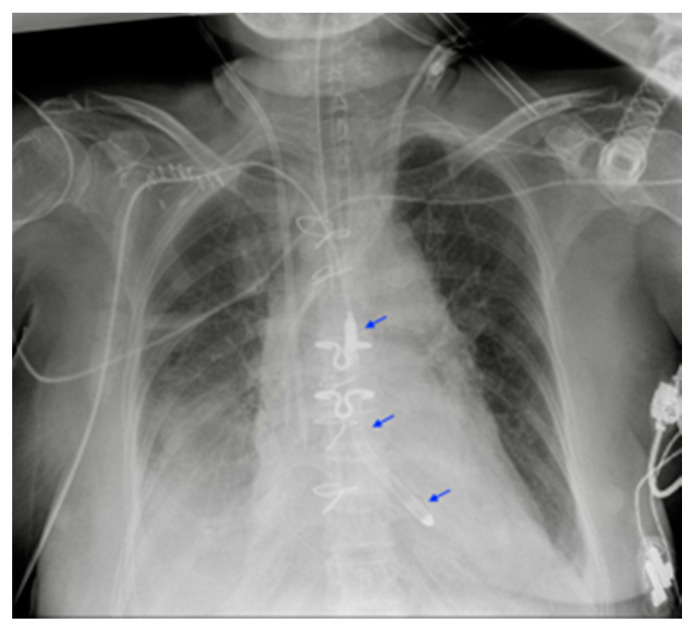
Impella in the correct position. Device is placed into the left ventricle and in the ascending aorta through the aortic valve (blue arrows).

**Figure 17 diagnostics-13-00599-f017:**
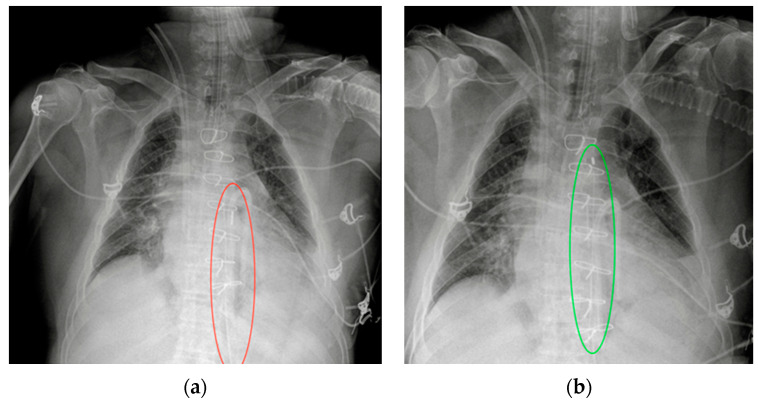
Malpositioned IABP. (**a**) Anteroposterior CXR of a patient with an IAPB placed too distal from the aortic arch: the upper radiopaque mark can be seen at the level of the sixth intercostal space (red circle). (**b**) The same patient after repositioning of the IABP, with the upper mark now just under the aortic arch, in the proximal thoracic descending aorta, at the level of the fourth intercostal space (green circle).

**Figure 18 diagnostics-13-00599-f018:**
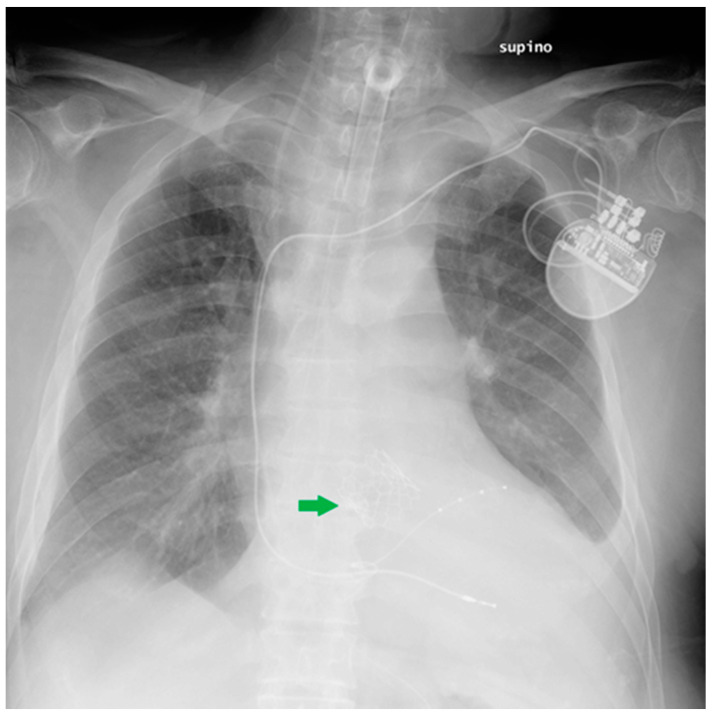
This CXR shows the result of a transcatheter aortic valve implantation (TAVI) (green arrow). The radiogram also includes a PM, a tracheostomy tube and a right jugular CVC.

**Figure 19 diagnostics-13-00599-f019:**
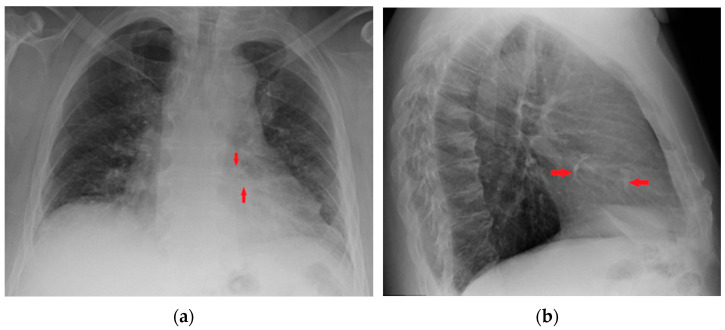
Coronary artery stents (red arrows), barely visible on the PA projection (**a**), can be better appreciated on the LL view (**b**).

**Figure 20 diagnostics-13-00599-f020:**
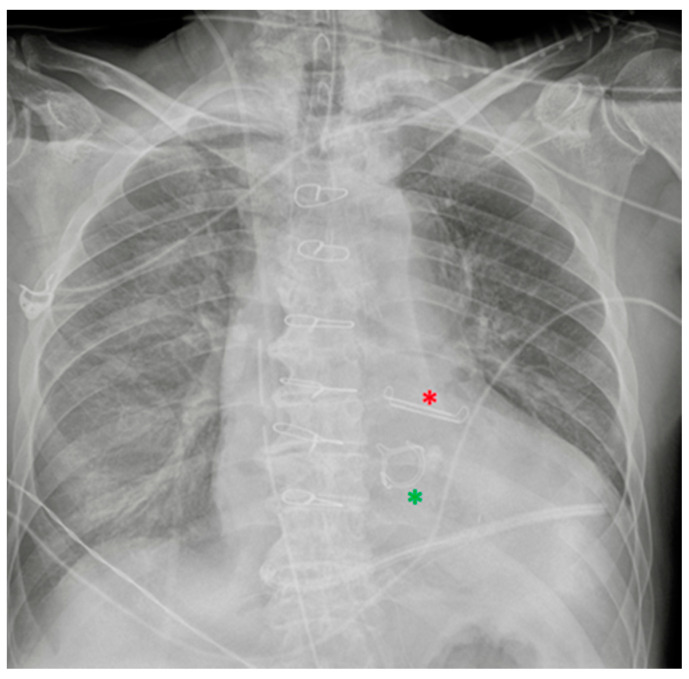
Left atrial appendage closure device (red asterisk). Prosthetic biological mitral valve (green asterisk). the image also shows right jugular CVC, ETT, NG-tube, chest tubes and sternal wires.

**Figure 21 diagnostics-13-00599-f021:**
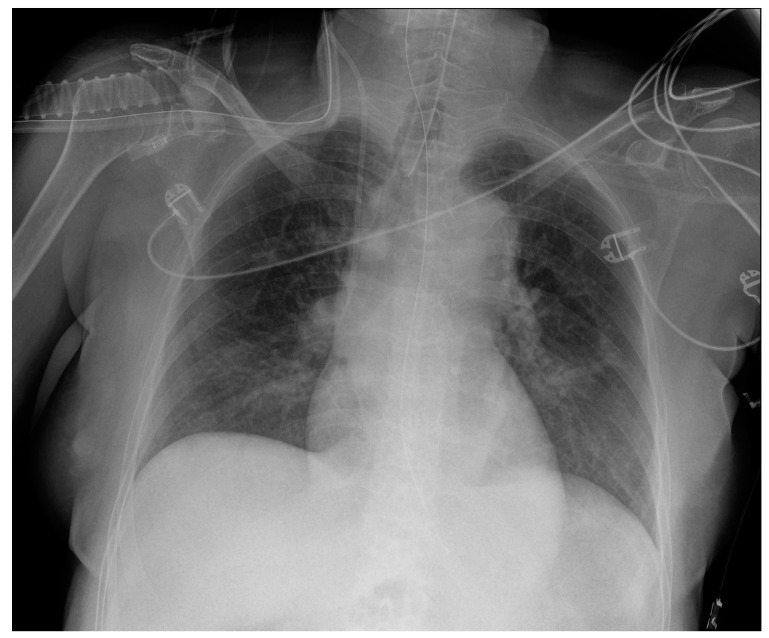
Image of a properly positioned nasogastric tube; also in the image, a well-positioned ETT.

**Figure 22 diagnostics-13-00599-f022:**
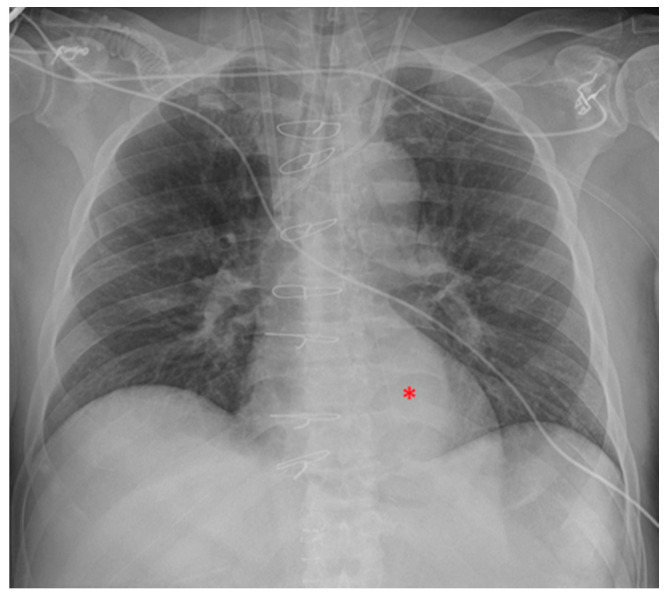
Image of NG tube positioned too proximally, its tip reaching the distal oesophagus and therefore projecting against the heart (red asterisk). In the same patient, the ETT has been pushed too low, with its tip proximal to the carina. The picture also shows two left jugular CVCs and a right jugular CVC.

**Figure 23 diagnostics-13-00599-f023:**
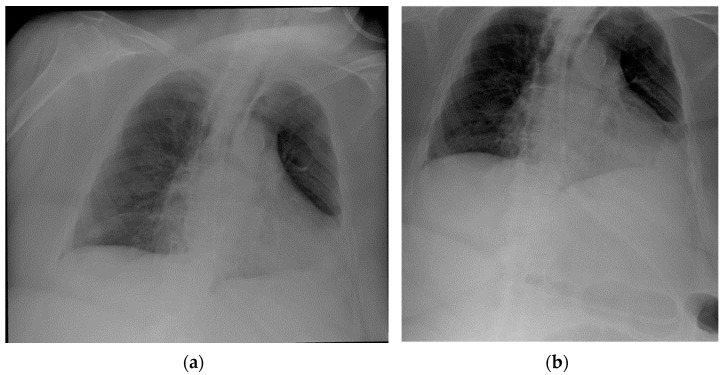
Images of a misplaced NG tube, whose tip reaches the pre-pyloric region after looping at the greater curvature of the stomach. Note that such a misposition was not visible in the regular chest projection (**a**), making an extended view on the upper abdomen therefore necessary (**b**).

**Figure 24 diagnostics-13-00599-f024:**
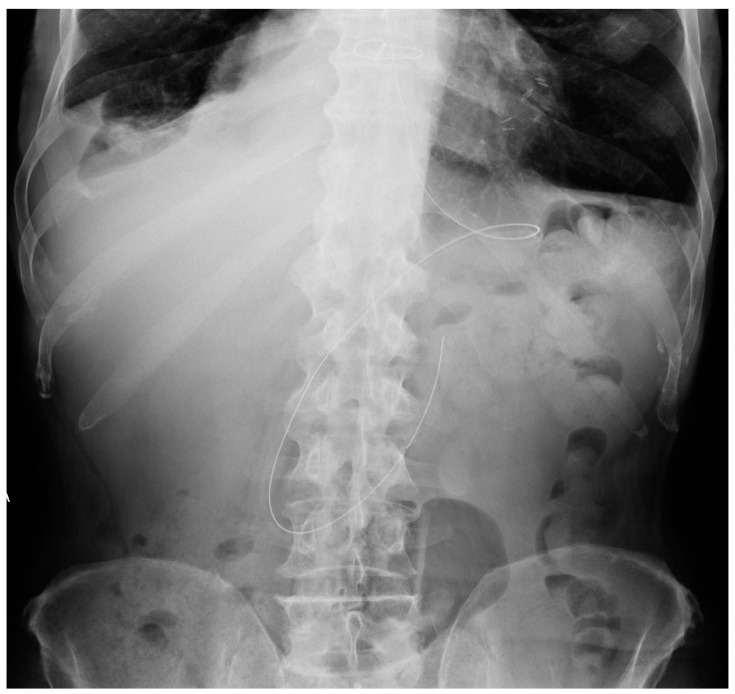
Abdomen X-ray showing the image of a NG tube that has been pushed too deep and reached the distal duodenum.

**Figure 25 diagnostics-13-00599-f025:**
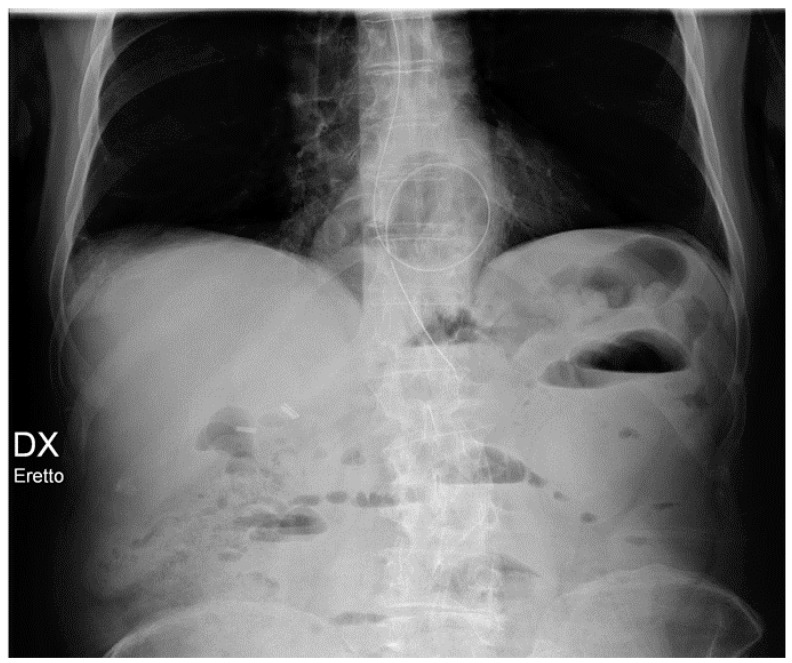
Abdomen X-ray showing the image of a NG tube projectively looping in the inferior mediastinum, due to a hiatal hernia; it is possible to also appreciate two metallic clips in the right hypochondrium and multiple air-fluid levels.

**Figure 26 diagnostics-13-00599-f026:**
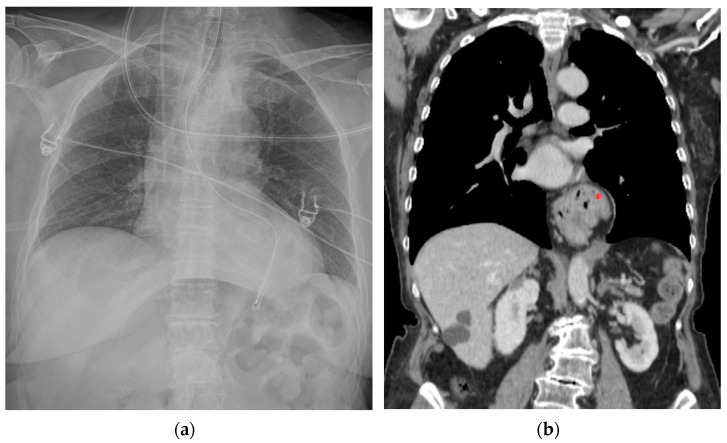
CXR of NG tube kinking shortly beneath the diaphragm, its tip projectively ending against the cardiac shadow (**a**); a right jugular CVC is also present. In the coronal view of the thoraco-abdominal CT scan of the same patient (before the NG tube was positioned) (**b**), it is possible to appreciate a hiatal hernia (red asterisk), which explicates the previous finding.

**Figure 27 diagnostics-13-00599-f027:**
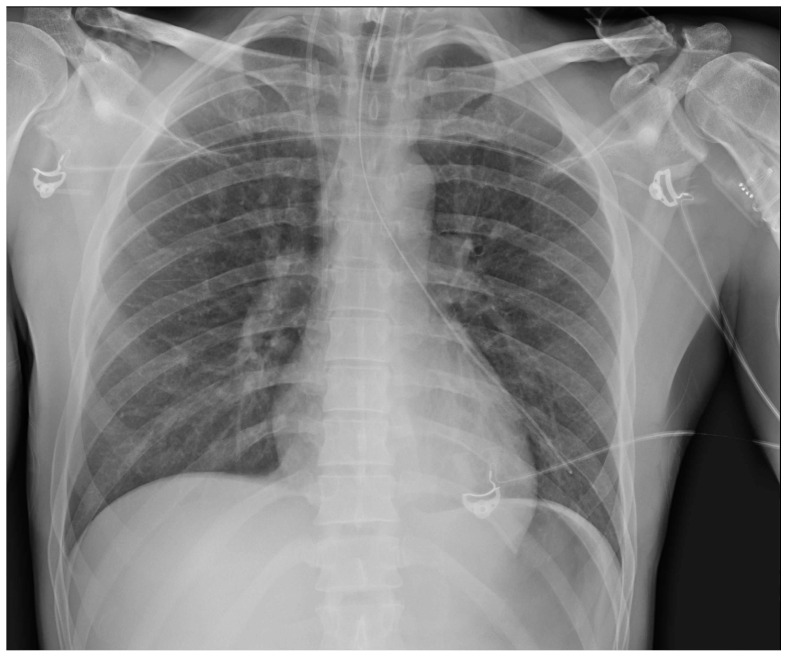
Image of a mispositioned NG tube, whose tip entered the left mainstem bronchus down to the left lower airways; prompt recognition of such misplacement avoided any sequelae for the patient. This CXR also shows a right jugular CVC and an ETT.

**Figure 28 diagnostics-13-00599-f028:**
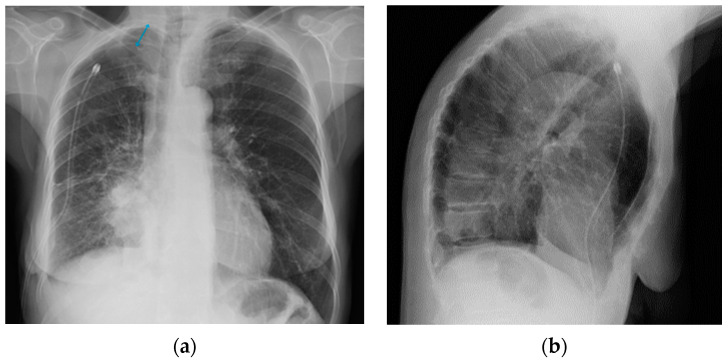
Right apical chest drainage tube in patient with right pneumothorax (blue arrow) on PA (**a**) and LL (**b**) projections.

**Figure 29 diagnostics-13-00599-f029:**
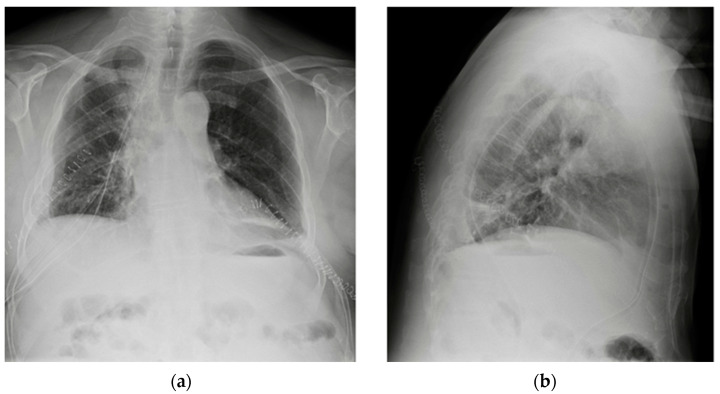
Post-operative CXR illustrating two chest tubes with their tip in the right apical region. Note that whereas in the PA projection (**a**), the tubes seem to follow the same course, the LL view (**b**) clearly differentiates between an anterior and a posterior course.

**Figure 30 diagnostics-13-00599-f030:**
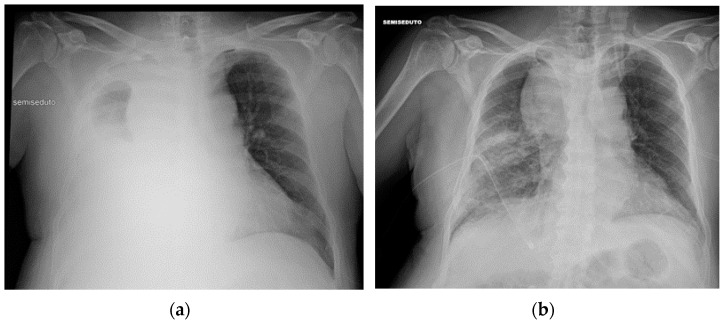
Images of a right pleural effusion before (**a**) and after (**b**) drainage through an inferior-posterior chest tube. Right enlargement of the upper mediastinum is also present.

**Figure 31 diagnostics-13-00599-f031:**
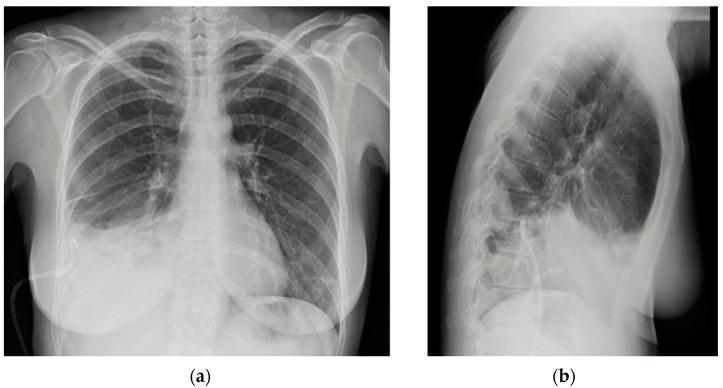
Image of a pig-tail chest tube inserted through the right lateral chest wall draining a right basal pulmonary effusion. Note that in this case, the lateral view (**b**) depicts the tube loops more clearly than PA view (**a**); also, there seems to be an effect given by the uplifting of the arms in the lateral view, with the insertion point of the tube being dragged upward and so “emphasizing” its looping appearance.

**Figure 32 diagnostics-13-00599-f032:**
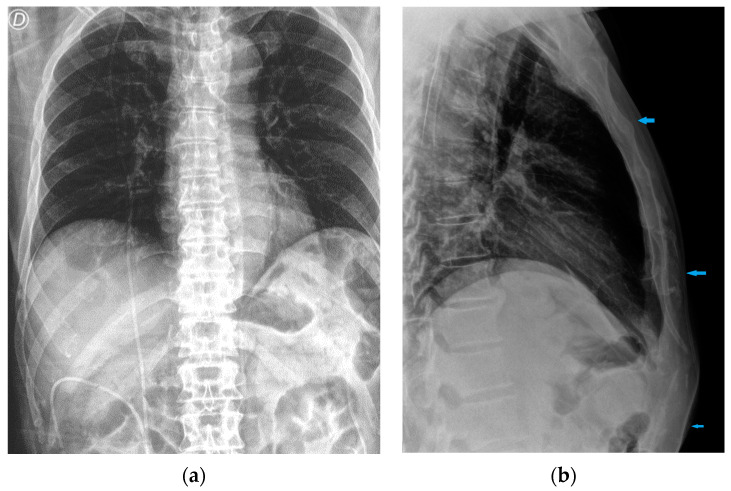
Image of the subcutaneous course (blue arrows) of a VP shunt on CXR, PA (**a**) and LL (**b**) views. When only the PA projection is available, it is important not to misinterpret the thoracic course of the VP shunt as a misplaced CVC.

**Figure 33 diagnostics-13-00599-f033:**
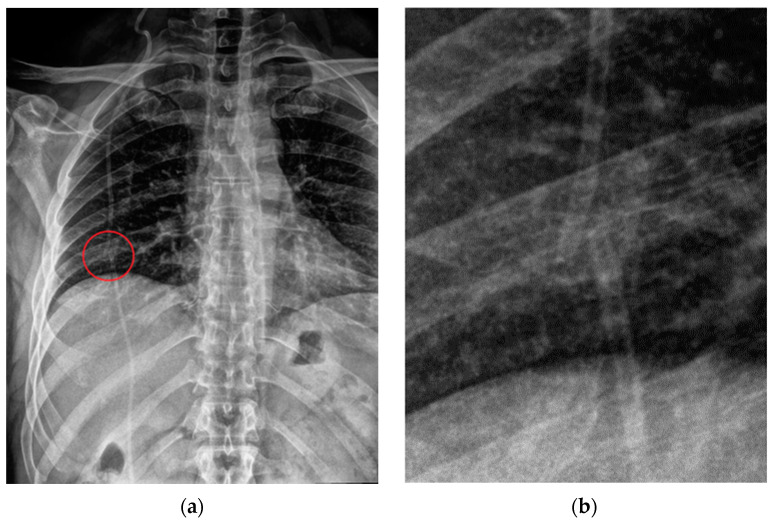
Rupture of a VP shunt (red circle) visible on the PA view of a CXR (**a**), magnified in the second image (**b**).

**Figure 34 diagnostics-13-00599-f034:**
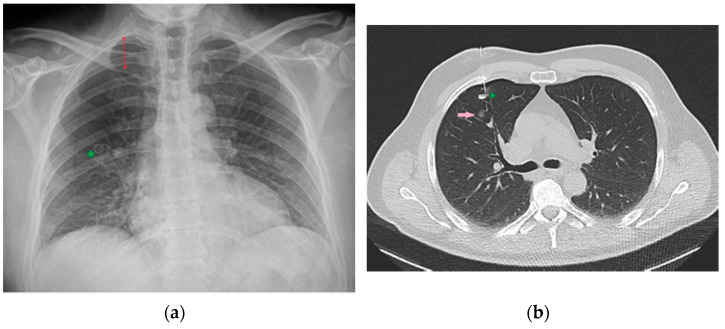
Image of a CXR (**a**) obtained after CT-guided positioning (**b**) of a microcoil (green asterisk) into the lung parenchyma as a landmark to more easily recognize and resect the adjacent nodule (pink arrow) intraoperatively. Development of pneumothorax as a complication following this procedure is not uncommon [36], as it is possible to appreciate this in the reported CXR (red double arrow).

**Figure 35 diagnostics-13-00599-f035:**
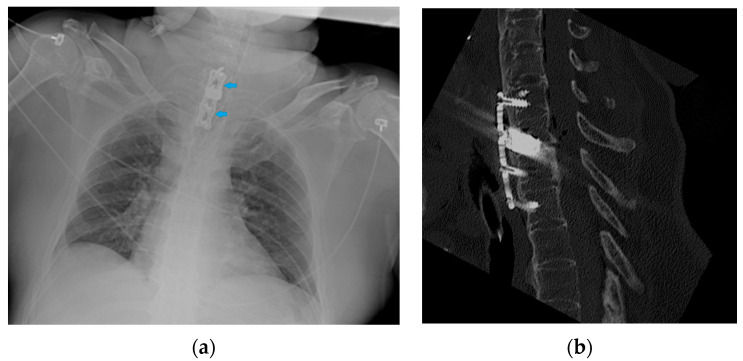
Image of an anterior spine stabilizer (blue arrows) on CXR (**a**) and the corresponding image on the CT scan (**b**).

**Figure 36 diagnostics-13-00599-f036:**
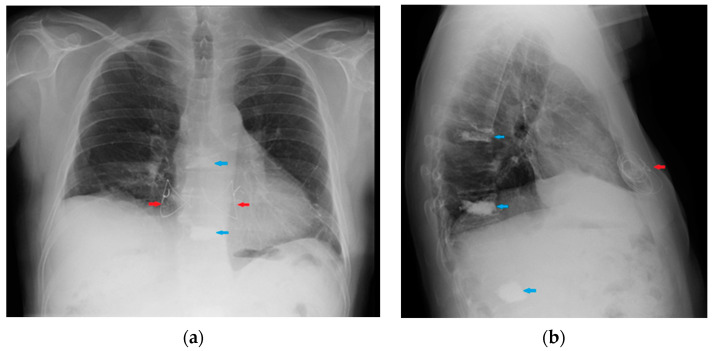
PA (**a**) and LL (**b**) CXR of a patient with sternal wires resulting from a transverse sternotomy (red arrows) and multiple outcomes of vertebroplasty (blue arrows).

**Figure 37 diagnostics-13-00599-f037:**
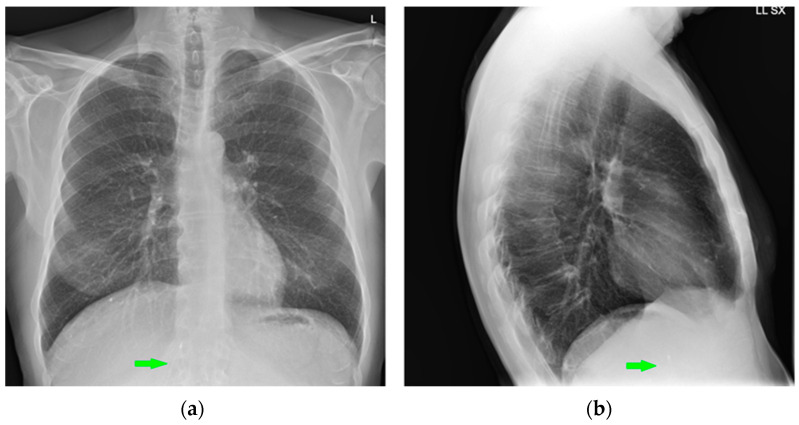
Image of an IVC filter on PA (**a**) and LL (**b**) views of a CXR (green arrows).

**Figure 38 diagnostics-13-00599-f038:**
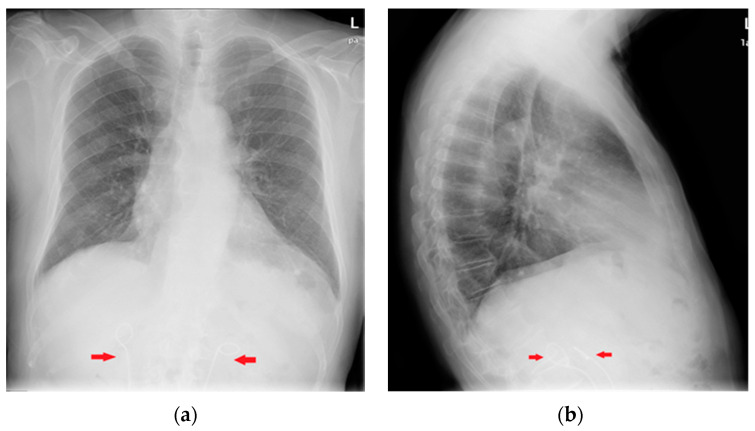
PA (**a**) and LL (**b**) CXR of a patient with bilateral ureteral stents (red arrows).

**Figure 39 diagnostics-13-00599-f039:**
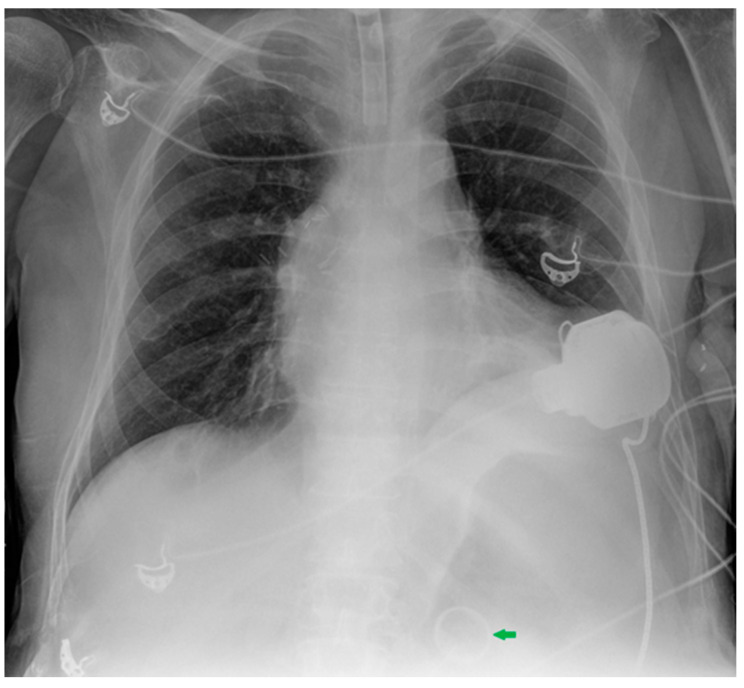
Image of a percutaneous endoscopic gastrostomy (PEG) (green arrow) on CXR. A tracheostomy tube and a LVAD are also present.

**Figure 40 diagnostics-13-00599-f040:**
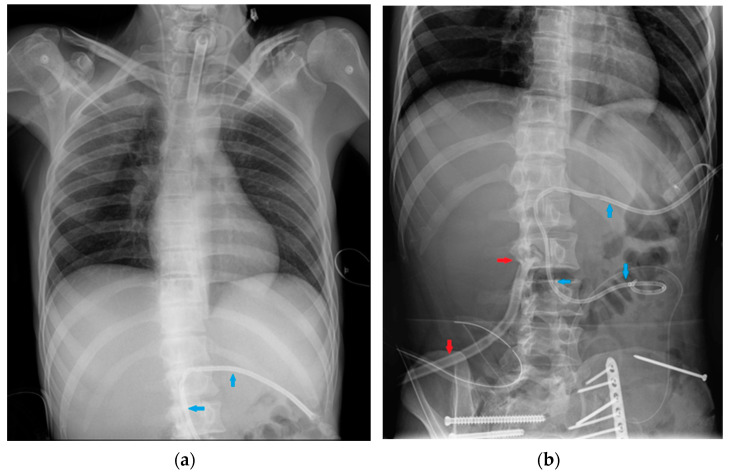
Image of a percutaneous transhepatic biliary drainage (PTBD) (blue arrows) partially included on the CXR (**a**) previously shown also in the Figure 4b (mispositioned left jugular CVC). The patient suffered a duodenum rupture after a car accident, so the PTBD was placed to drain the bile from the choledocum into the distal part of the duodenum, therefore bypassing the point of rupture, as shown in the abdominal X-ray (**b**); a Pezzer catheter (red arrows) was also placed in the duodenal lumen through the breach to further drain eventual other secretions. Multiple other drainage tubes as well as metallic screws and plate are present.

**Table 1 diagnostics-13-00599-t001:** Anatomical-based classification of chest devices.

Device Classification	
Intravascular devices	Central vein catheter
	Peripherally inserted central catheter
	Port Swan-Ganz catheter
Airway devices	Endotracheal tubes
	Tracheostomy tubes
Cardiac devices	Loop recorder
	Pacemaker and automatic implantable cardioverter-defibrillator Ventricular assistance devices
	Impella
	Intra-aortic balloon pump Other cardiac devices
Gastro-intestinal devices	Nasogastric tubes
	Nasoenteric tubes
Chest tubes	

Miscellaneous	


**Table 2 diagnostics-13-00599-t002:** Intravascular devices’ most frequent CXR-detectable anomalies and complications.

Device	Most Frequent CXR-Detectable Anomalies and Complications
CVC	Malposition (looping, IVC tip positioning, wrong thoracic vein cannulation)
	Pneumothorax
	Vascular perforation (less frequent)
PICC	Misposition
	Pneumothorax/hemothorax (infrequent)
PORT	Pneumothorax
Midline	Misposition
Swan-Ganz catheter	Malposition (looping, knotting)
	RA or pulmonary artery wall perforation (diagnosis mostly clinical- or echocardiography-based)

**Table 3 diagnostics-13-00599-t003:** Airway devices’ most frequent CXR-detectable mispositions and respective complications.

Device	CXR-Detectable Mispositions	Complications
ETT	Laryngeal tip positioning	Cord trauma
	Bronchus cannulation	Contralateral atelectasis, ipsilateral overinflation (risk of barotrauma, pneumothorax)
	Oesophageal intubation	Non-improving respiratory function, esophageal perforation
	Balloon overinflation	Throat pain, tissue ischemia, tracheal perforation, fistulae formation
Tracheostomy tube		Subcutaneous emphysema Hematoma Pneumomediastinum Tracheal stenosis (late complication)

**Table 4 diagnostics-13-00599-t004:** Cardiac devices’ most frequent CXR-detectable anomalies and complications.

Device	Most Frequent CXR-Detectable Anomalies and Complications
Loop recorder	Migration (anecdotical)
PM and AICD	Leads misposition/breakage
	Twiddler’s syndrome
VADs	Hemopericardium, hemothorax
Impella	Dislocation
IABP	Misposition
ECMO	Misposition

**Table 5 diagnostics-13-00599-t005:** Intravascular devices’ most frequent CXR-detectable anomalies and complications. * Additional X-ray series/different imaging techniques may be required for detection.

Device	Most Frequent CXR-Detectable Anomalies and Complications
Nasogastric tubes	Proximal (oesophagus)/distal (duodenum) * misposition Coiling/kinking Insertion into the respiratory tree → aspiration pneumonia Gastric wall perforation Intracranial insertion (rare) *
Nasoenteric tubes	Comparable to NG tubes

**Table 6 diagnostics-13-00599-t006:** Chest tubes’ most frequent CXR-detectable anomalies and complications.

Device	Most Frequent CXR-Detectable Anomalies and Complications
Chest tubes	Kinking Extrapleural/intrafissural/intraparenchymal/misposition Mediastinum juxtaposition Diaphragmatic trespassing Mediastinal invasion (uncommon)

**Table 7 diagnostics-13-00599-t007:** CSF shunts’ most frequent CXR-detectable anomalies and complications.

Device	Most Frequent CXR-Detectable Anomalies and Complications
CSF shunts	Breakages Disconnection/migration of the distal catheter Pneumothorax Subcutaneous emphysema Pulmonary hypertension (VA shunts only)

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
