# Peer review of "Chest X-ray Interpretation: Detecting Devices and Device-Related Complications"

_diagnostics, 2023, doi:10.3390/diagnostics13040599_

Round 1

Reviewer 1 Report

I have reviewed the manuscript “Chest X-Ray interpretation: detecting devices and device-related complications”, which provides a useful revision of commonly encountered medical devices during chest x-ray reading.

-          In section 2. Classification of Medical Devices, please add a figure to illustrate the proposed classification, which is further developed in the following sections.

-          At the end of each of subsection within sections 3-8, please provide a Figures to summarize device-related complications.

Author Response

Dear Reviewer,

Thank You very much for your review.

- As you suggested, in the latest revision we added a table at the end of section 2 (Classification of Medical Devices) which further explicates the classification used for chest devices.

- We also added tables at the end of each section resuming the most frequent anomalies and complications detectable on CXR for the respective group of devices.

Best regards,

The Authors

Reviewer 2 Report

Dear Authors,

this is a well-written manuscript presenting a very interesting diagnostic issue, which is the detection of devices and device-related complications in chest X-ray. Please pay attention to the following comments and queries pertaining to your manuscript:

1.      Line 45: please correct us such: in critically ill patients.

2.      Lines 105-106: please correct us such: these are life threatening situations that usually require surgical treatment and the diagnosis is mainly based on clinical features or bedside echocardiographic findings.

3.      Regarding the NG tube placement, please note that often an abdomen X-ray (like seen for example in Figures 22 and 23) may be additionally required in order to define the abdominal position of the tube.

4.      Please add the following devices which can be detected with chest X-ray: right and left ventricular assist devices (RVAD, LVAD), ECMO-Cannula, temporary pacemaker.

5.      Please refer that all provided X-rays have been anonymized.

With Best Regards

Author Response

Dear Reviewer,

Thank You for your review.

We entirely fullfilled your requests at point 1, 2, 3 and 5.

Sections describing VADs and ECMO have been added in the "Cardiac Devices" section; a mention to temporary PM has been added to the relative sub-section (PMs and AICDs).

Best regards,

The Authors